# Enhanced Epithelial-to-Mesenchymal Transition and Chemoresistance in Advanced Retinoblastoma Tumors Is Driven by miR-181a

**DOI:** 10.3390/cancers14205124

**Published:** 2022-10-19

**Authors:** Vishnu Suresh Babu, Anadi Bisht, Ashwin Mallipatna, Deepak SA, Gagan Dudeja, Ramaraj Kannan, Rohit Shetty, Nilanjan Guha, Stephane Heymans, Arkasubhra Ghosh

**Affiliations:** 1GROW Research Laboratory, Narayana Nethralaya Foundation, Bangalore 560099, India; 2Department of Cardiology, Cardiovascular Research Institute Maastricht (CARIM), Maastricht University, 6229 ER Maastricht, The Netherlands; 3Retinoblastoma Service, Narayana Nethralaya, Bangalore 560099, India; 4Agilent Technologies India Pvt. Ltd., Bangalore 560048, India; 5Centre for Molecular and Vascular Biology, Department of Cardiovascular Sciences, Katholieke Universiteit Leuven, Herestraat 49, bus 911, 3000 Leuven, Belgium

**Keywords:** EMT, miRNA, retinoblastoma, chemo-resistant

## Abstract

**Simple Summary:**

Our study identified the differential expression and potential effects of microRNAs in retinoblastoma vs. pediatric retina and advanced vs. non-advanced tumors. We provide evidence of the epithelial–mesenchymal transition (EMT) and chemoresistance programs in advanced tumors, which were potentially attributed to miR-181a-5p. We analyzed the differential expression of relevant EMT- and chemoresistance-related proteins in advanced vs. non-advanced tumors and chemotherapy-adapted Y79 cells to assess whether EMT and chemoresistance mechanisms were linked. We further examined the possible role of TGFβ as a potential regulator of such differences and highlighted the role of miR-181a-5p in EMT- and chemoresistance-related gene expression and drug sensitivity.

**Abstract:**

Advanced retinoblastoma (Rb) tumors display high metastatic spread to distant tissues, causing a potent threat to vision and life. Through transcriptomic profiling, we discovered key upregulated genes that belonged to the epithelial–mesenchymal transition (EMT) and chemotherapy resistance pathways in advanced Rb tumors. Through in vitro models, we further showed that Rb null tumor cells under prolonged chemo drug exposure, acquires a metastasis-like phenotype through the EMT program mediated by ZEB1 and SNAI2 and these cells further acquires chemotherapeutic resistance through cathepsin-L- and MDR1-mediated drug efflux mechanisms. Using a miRNA microarray, we identified miR-181a-5p as being significantly reduced in advanced Rb tumors, which was associated with an altered EMT and drug-resistance genes. We showed that enhancing miR-181a-5p levels in Rb null chemo-resistant sublines reduced the ZEB1 and SNAI2 levels and halted the mesenchymal transition switch, further reducing the drug resistance. We thus identified miR-181a-5p as a therapeutically exploitable target for EMT-triggered drug-resistant cancers that halted their invasion and migration and sensitized them to low-dose chemotherapy drugs.

## 1. Introduction

Retinoblastoma (Rb) is the most common intraocular malignant tumor in children. Managing intraocular Rb tumors via efficient diagnoses, genetic screening, and clinical procedures [1,2] help to achieve excellent survival rates worldwide. However, metastatic retinoblastoma is still a major concern in many countries [3,4,5]. Rb tumors that grow rapidly have sufficient feeder arteries and drainage veins and are characterized by the presence of a multifocal yellowish-white tumor mass with floating subretinal or vitreous cancer seeds [6]. If neglected or untreated, advanced Rb tumors demonstrate massive choroidal invasion [7] and metastatic spread, primarily through the optic nerve [8] and sclera [9], to regional lymph nodes, the central nervous system (CNS), and bone marrow [10], causing a potent threat not only to vision but to the life of the child. To manage metastatic Rb tumors clinically, an intensive multimodal approach that incorporates high-dose systemic, intra-arterial, and peri-orbital chemotherapy regimens involving carboplatin, etoposide, and cyclophosphamide followed by radiation is currently used [11]. However, advanced tumors often evolve during successive chemotherapy cycles and develop resistance to anticancer therapeutics, diminishing the efforts of the clinical management procedures [12,13]. Upon prolonged chemo-drug exposure, advanced Rb tumors increase the expression of ATP binding cassette (ABC) transporter pathway genes, such as MDR1 and MRP1, to confer resistance via a chemo-drug efflux mechanism [14]. Metastatic tumors acquire chemotherapy resistance through trans-differentiation that is initiated by the epithelial-to-mesenchymal transition (EMT) program in different cancers [15,16]. The EMT program begins with the loss of epithelial phenotypes via the downregulation of E-cadherin and tight junction adhesion molecules. The differentiated cancer cells change to the mesenchymal phenotype with an invasive dedifferentiated characteristic, which can coincide with acquiring chemo-drug-resistance properties.

MicroRNAs (miRNA) are small non-coding single-strand RNAs that emerged as an important modifier of a plethora of biological pathways, including for cancers [17]. They modify gene expression by using the RNA-induced silencing complex (RISC) that binds to the 3′ untranslated region (UTR) or, less frequently, the 5′ UTR region of the mRNA and cause translational repression. Emerging evidence points out the role of miRNAs in controlling EMT transcription factors and signaling pathways to regulate metastatic dissemination in different cancers [18]. In Rb tumors, the increased expression of the miR17-92 cluster [19], miR-25-3p [20], and miR200c [21] were found to regulate high EMT-mediated invasion and migration of Rb cells in vitro, thus supporting the role of EMT in Rb metastasis. However, the mechanistic links between miRNAs, the EMT, and drug resistance in Rb tumors remain obscure.

In the present study, we profiled miRNA and mRNA signatures simultaneously in the same set of advanced and non-advanced Rb tumors and compared the results with those of age-matched healthy pediatric retinae. Such a coordinated analysis of expression networks in the same set of tissues and controls enabled the discovery of co-regulated miRNA and mRNA targets relevant to the Rb stage. Among the many dysregulated genes and miRNAs, we chose to validate and investigate the functional role of miR-181a-5p on the enhanced EMT and drug resistance pathways in advanced Rb subjects.

## 2. Materials and Methods

### 2.1. Clinical Samples

This study complied with the Declaration of Helsinki and was performed according to a protocol approved by the institutional ethics committee of Narayana Nethralaya (EC ref no: C/2013/03/02). Written informed consent was received from all parents of the subjects before inclusion in the study. After histopathological examination, Rb tumors (*n* = 9) were divided into group E and group D of the age range 0.2–4 years, and pediatric controls (*n* = 2) of the age range (0.2–0.3 years) were used for the miRNA and mRNA microarray study. The details of clinical samples, including age, gender, laterality, tumor viability, clinical, and histopathology details are mentioned in Table 1. For the RT-PCR validations, we used additional Rb subjects comprising group E (*n* = 4), group D (*n* = 4), and pediatric retina (*n* = 4) of the age range 0.2–4 years. The clinical and histopathology details of additional Rb subjects are mentioned in Appendix A. For the immunohistochemistry validations, we used additional Rb subjects comprising group E (*n* = 12), group D (*n* = 12), and pediatric retina (*n* = 4). The clinical and histopathology details of the additional Rb subjects are mentioned in Appendix A.

### 2.2. Tumor miRNA and mRNA Profiling

Total RNA was isolated from 9 Rb tumors and 2 control pediatric retina samples using an Agilent Absolutely RNA miRNA kit (cat# 400814, Agilent Technologies, Santa Clara, CA, USA) according to the manufacturer’s instructions. The quality of the isolated RNA was determined on an Agilent 2200 TapeStation system (cat#G2964AA, Agilent Technologies, Santa Clara, CA, USA) using an Agilent RNA ScreenTape assay (cat#5067-5576, Agilent Technologies, Santa Clara, CA, USA). mRNA labeling and microarray processing was performed as detailed below in the “One-Color Microarray-Based Gene Expression Analysis” (cat# G4140-90040, Agilent Technologies, Santa Clara, CA, USA). miRNA labeling was done using an Agilent miRNA Complete Labeling and Hybridization Kit (Cat# 5190-0456, Agilent Technologies, Santa Clara, CA, USA). The gene expression and miRNA data were extracted using Agilent Feature Extraction Software (11.5.1.1) and analyzed using Agilent GeneSpring GX 13.1. The analysis was carried out using a *t*-test unpaired statistical method with the Benjamini–Hochberg FDR method. In both the mRNA and miRNA analyses, transcripts exhibiting *p* ≤ 0.05 and fold changes greater than or equal to two were considered differentially expressed entities. Both the mRNA (GSE208143) and miRNA (GSE208677) microarray data were submitted to the NCBI GEO database.

### 2.3. Cell Lines

Y79 and WERI-Rb1 cells were obtained from the American Type Culture Collection (ATCC, Manassas, VA, USA). The Y79 and WERI-Rb1 cells were cultured in RPMI 1640 medium (Cat #11875093, Gibco, Grand Island, NY, USA) supplemented with 10% FBS (cat#A4766801, Gibco, Grand Island, NY, USA) and 1% Pen Strep (Penicillin–Streptomycin) (cat#15070063, Gibco, Grand Island, NY, USA) and maintained at 37 °C in a humidified atmosphere of 5% CO_2_ with intermittent shaking in an upright T25 flask. To generate chemotherapy-resistant lines, Y79 and WERI-Rb1 cells were exposed to media containing a low dose (1/100th of the IC_50_) of topotecan (cat#1672257, Sigma Aldrich, St. Louis, MO, USA) or carboplatin (cat#216100-M, Sigma Aldrich, St. Louis, MO, USA) for 48 h and replenished with fresh media without drugs for the next 48 h and vice versa. At the end of each week, we increased the dose of topotecan and carboplatin by 10-fold for 3–4 weeks till the cells displayed tight large clusters and no sensitivity to chemo-drugs. The cells were further analyzed for their MDR1 surface expression and IC_50_ shift to confirm the resistant phenotype.

### 2.4. Gene Expression Analysis

Total RNA extracted from the second cohort of clinical subjects was used for the RT PCR validation for the mRNA and miRNA microarray. RT-PCR was performed with Agilent Brilliant III Ultra-Fast RT-PCR reagent (cat# 600884, Agilent Technologies, Santa Clara, CA, USA) using Agilent AriaMX real-time PCR instruments. Relative mRNA expression was quantified using the ∆∆C(t) method. For in vitro assays, total RNA was isolated from cells using the Trizol reagent (Invitrogen, Carlsbad, CA, USA) according to the manufacturer’s protocol. Then, 1 µg of RNA was reverse-transcribed using a Bio-Rad iScript cDNA synthesis kit (cat# 1708890, Bio-Rad, Hercules, CA, USA) and quantitative real-time PCR was performed using a Kappa Sybr Fast qPCR kit (cat# KK4601, Kapa Biosystems Pty Ltd., Wilmington, MA, USA) using a Bio-rad CFX96 system. Relative mRNA expression levels were quantified using the ∆∆C(t) method. Results were normalized to housekeeping human β-actin. Details of the primers used are described in Table 2 below.

For the qPCR of the miRNAs, miRNA was converted to cDNA using the miRCURY LNA Universal RT microRNA PCR reverse transcription kit (cat#339306, Qiagen, Hilden, Germany). Briefly, RNA was polyadenylated with ATP using poly(A) polymerase at 37 °C for 1 h and reverse-transcribed using 0.5 μg of poly(T) adapter primer. Each miRNA was detected by the mature DNA sequence as the forward primer and a 3′ universal reverse primer provided in the QuantiMir RT kit (cat#RA420A-hU6, System Biosciences, Palo Alto, CA, USA). Human small nuclear U6 RNA was amplified as an internal control. The qPCR was performed using Power SYBR Green PCR Master Mix (cat# 4367659, Applied Biosystems, Waltham, MA, USA). All qPCRs were performed using SYBR Green and were conducted at 50 °C for 2 min, 95 °C for 10 min, and then 45 cycles of 95 °C for 10 s and 60 °C for 1 min. The specificity of the reaction was verified via a melt curve analysis. The details of the miRNA primers used are mentioned in Table 3.

### 2.5. Histopathology and Light Microscopy

Paraffin-embedded specimens of the Rb tumors (*n* = 9) and control retinas (*n* = 2) were used. Four-micrometer paraffin sections were dewaxed at 60 °C and rehydrated in decreasing concentration of ethanol. Slides were stained with hematoxylin and eosin according to standard procedures. Brightfield images were captured using an Olympus CKX53 microscope.

### 2.6. Immunofluorescence

For the IF analysis, 4 µm sections of Rb tumors (*n* = 25) and pediatric retinas (*n* = 2) were deparaffinized, rehydrated, and subjected to heat-induced epitope retrieval using a citrate buffer for 20 min at 100 °C. After a 2% BSA block, tissues were incubated overnight at 4 °C with antibodies for ZEB1 (1:1000; cat#70512, Cell Signaling Technology, Danvers, MA, USA), Cathepsin L (1:500; cat#ab6314, Abcam, Cambridge, UK), E-cadherin (1:500; cat#3195, Cell Signaling Technology, Danvers, MA, USA), N-cadherin (1:1000; cat#13116, Cell Signaling Technology, Danvers, MA, USA), and MDR1 (1:1000, cat#13342, Cell Signaling Technology, Danvers, MA, USA). For the in vitro experiments, 2 × 10^3^ parental and topotecan-resistant Y79 cells were seeded on 8-chamber glass slides that were precoated with 0.01% poly-L-lysine (cat#P4707, Sigma Aldrich, St. Louis, MO, USA). The cells were stained with phospho-λ-H2A.x (ser139) (1:500; cat#9718, Cell Signaling Technology, Danvers, MA, USA), ZEB1 (1:500; cat#70512, Cell Signaling Technology, Danvers, MA, USA), Cathepsin L (1:500; cat#ab6314, Abcam, Cambridge, UK), MDR1 (1:500, cat#13342, Cell Signaling Technology, Danvers, MA, USA), phospho-SMAD2 (1:1000, cat#ab53100, Abcam, Cambridge, UK), and TGFBR2 (1:1000, cat#ab78419, Abcam, Cambridge, UK). Secondary antibodies used included goat anti-mouse Alexa Fluor 488 (1:5000, cat# ab150113, Abcam, Cambridge, UK) and donkey anti-rabbit Cy3 (1:5000, cat# 711-1650152, Jackson ImmunoResearch Laboratories, West Grove, PA, USA). Hoechst 33342 (1:5000, cat#H1399, Invitrogen, Waltham, MA, USA) was used for nuclear staining. Images were analyzed and captured using EVOS M7000 imaging systems (ThermoFisher Scientific, Waltham, MA, USA). The fluorescent intensity was measured using ImageJ software (NIH Image, Bethesda, MD, USA).

### 2.7. Western Blotting

For the Western blot analysis, cells were lysed in a RIPA buffer (recipe: 20 mM Tris pH 8.0, 0.1% SDS, 150 mM NaCl, 0.08% sodium deoxycholate, 1% NP40 supplemented with 1 tablet of protease inhibitor (Complete ultra mini-tablet, Roche)), and 1 tablet of phosphatase inhibitor (PhosphoStop tablet, Roche, Basel, Switzerland) for 30 min on ice. Total protein (20 µg) was loaded per lane and was separated using SDS-PAGE. The separated proteins on the gel were transferred onto a PVDF membrane and were probed for specific antibodies against Rb (cat# 9309; Cell Signaling Technology, Danvers, MA, USA) phospho-Rb (cat# 8516 Cell Signaling Technology, Danvers, MA, USA), E2F1 (1:500, cat#sc193, SantaCruz Biotechnology, Dallas, TX, USA), ZEB1 (1:1000; cat#3396, Cell Signaling Technology, Danvers, MA, USA), Cathepsin L (1:1000; cat#ab6314, Abcam, Cambridge, UK), Slug (1:1000; cat#9585, Cell Signaling Technology, Danvers, MA, USA), E-cadherin (1:1000; cat#3195, Cell Signaling Technology, Danvers, MA, USA), N-cadherin (1:1000; cat#13116, Cell Signaling Technology, Danvers, MA, USA), MDR1 (1:1000, cat#13342, Cell Signaling Technology, Danvers, MA, USA), Total smad2/3 (1:1000, cat#ab207447, Abcam, Cambridge, UK), phospho-SMAD2 (1:1000, cat#ab53100, Abcam, Cambridge, UK), TGFBR1 (1:1000, cat#PA1731, BosterBio Pleasanton, CA, USA), TGFBR2 (1:1000, cat#ab78419, Abcam, Cambridge, UK), α-Tubulin (1:1000, cat# 3873; Cell Signaling Technology, Danvers, MA, USA), and GAPDH (1:1000, cat#5174; Cell Signaling Technology, Danvers, MA, USA) in 5% BSA in 1X TBST overnight at 4 °C. For the nuclear-cytoplasmic fractionation, the cytoplasmic fraction was extracted using a hypotonic buffer for 30 min on ice and the nuclear fraction was extracted using a lysis buffer solution containing 10 mM Tris at pH 8, 170 mM NaCl, and 0.5% NP40 with protease inhibitors. The respective cellular fractions were incubated with respective primary antibodies for immunoprecipitations. Lamin A/C (1:1000, sc-6215, Santa Cruz Biotechnology, Dallas, TX, USA) was used as a nuclear-fraction-loading control and α-Tubulin was used as a cytoplasmic-fraction-loading control (1:1000, cat# 3873; Cell Signaling Technology, Danvers, MA, USA). After 4 washes with 1X TBST for 10 min, membranes were incubated with HRP-conjugated anti-mouse (cat#7076, Cell Signaling Technology, Danvers, MA, USA) or anti-rabbit antibodies (cat#7074, Cell Signaling Technology, Danvers, MA, USA) at 1:2000 dilution for 2 h. Images were visualized using the Image Quant LAS 500 system (GE Healthcare Life Sciences, Piscataway, NJ, USA).

### 2.8. FACS Analysis of MDR1 Surface Staining

The cell surface expression of MDR1 in parental and resistant Y79 and WERI-Rb1 cells was detected using an anti-MDR1 antibody (1:500, cat#13342, Cell Signaling Technology, Danvers, MA, USA). Parental and resistant Y79 and WERI-Rb1 cells after the drug exposure were incubated in 200 µL of PBS containing 1% FBS and 2 µg of MDR1 antibody at 4 °C for 1 h in an intermittent shaker. After three washes with ice-cold PBS, the cells were further incubated in goat anti-rabbit Alexa Fluor 488 secondary antibody for 30 min at RT. The cells were then washed in ice-cold PBS and analyzed with a FACS apparatus equipped with FACSDiva software. The fluorescent intensity of the FL1 channel was plotted to compare the cell surface expression of MDR1 in parental and resistant lines.

### 2.9. Cell Proliferation Assay

Parental Y79 and WERI-Rb1, topotecan-resistant Y79 and WERI-Rb1, carboplatin-resistant Y79, and WERI-Rb1 cells were used for the proliferation assay. A total of 10,000 cells were seeded in 24-well plates for the proliferation assay. Cell viability was determined once every 24 h for 4 consecutive days using trypan blue cell staining and cell counting using a hemocytometer. In the miRNA-transfected models, 10,000 cells were seeded onto 24-well plates after 48 h of transfection, and proliferation was assessed from 24 h to 96 h. The cell viability was determined using a trypan blue assay. The experiments were performed in three experimental repeats in triplicates for different experimental conditions. Data were expressed as the mean  ±  SD of triplicate experiments.

### 2.10. Cell Migration and Invasion Assays

Cell migration and invasion assays were performed in 24-well transwell plates with cell culture inserts (BD Falcon). A total of 15,000 parental and resistant Y79 or WERI-Rb1 cells in 150 μL 0% RPMI media were seeded in a transwell insert coated with 1% matrigel and incubated for 48 h. The bottom chamber was filled with 600 μL of 10% RPMI media. After 48 h of incubation, cells on the insert were removed using a cotton swab. Migrated cells on the lower surface of the insert membrane were fixed with 4% PFA and stained with 0.1% crystal violet. Images were captured in a bright field using an Olympus CKX53 microscope. Cells were further lysed using 10% SDS and the absorbance of crystal violet was measured at 595 nm using a microplate reader. For the migration assay, the cells that migrated to the bottom chamber at 48 h were counted using trypan blue cell staining and cell counting using a hemocytometer.

For the miRNA transfection experiments, 15,000 topotecan-resistant Y79 and WERI-Rb1 cells were seeded in 0% RPMI media in the transwell insert coated with 1% matrigel for 48 h. Invasive and migrated cells were quantified using a 0.1% crystal violet staining protocol. Data were expressed as replicate data points ± SD of triplicate experiments.

### 2.11. Colony Formation/Tumor Spheroid Assay

The spheroid formation assays were carried out on a low-attachment U-bottom 96-well plate (BRAND^®^ 96-well microplate, Sigma Aldrich, St. Louis, MO, USA). A single-cell suspension of 500 parental and topotecan-resistant Y79 cells in 10% RPMI medium was loaded in each well of a 96-well plate followed by centrifugation at 1000 rpm for 1 min to facilitate cell aggregation. The cells were cultured at 37 °C in a 90% humidified incubator with 5% CO_2_ for 7 days for the generation of tight and regular tumor spheroids. Spheroids were imaged using the EVOS FL imaging system (ThermoFisher Scientific, Waltham, MA, USA). ImageJ 2.1 software was used for spheroid area measurements. Data were expressed as replicate data points ± SD of triplicate experiments.

### 2.12. Chemosensitivity Assay

The cell viability of topotecan-resistant Y79 and WERI-Rb1 cells after the miRNA transfections and exposure to topotecan IC_50_ (10 nm) treatment for 48 h was determined using the Presto Blue cell viability reagent (Invitrogen) as per the manufacturer’s protocol. In brief, topotecan-resistant Y79 or WERI-Rb1 (5 × 10^3^) were plated into 96-well plates (Eppendorf, Sigma Aldrich, St. Louis, MO, USA) and incubated overnight. The cells were treated with topotecan IC_50_ for 48 h. Untreated Y79 or WERI-Rb1 resistant cells were considered as the control. Four hours before the end of the treatment, presto-blue reagent (Invitrogen) was added and incubated for 2 h followed by measurement of fluorescence (540 nm excitation/590 nm emissions). The chemosensitivity of all treated cells was determined across conditions and compared against control mock-treated cells (which were considered 100% viable). Data were expressed as the mean ± SD of triplicate experiments.

### 2.13. miRNA–mRNA Target Prediction and Network Analysis

The miRNA–mRNA target prediction was performed using the databases miWalk 2.0, miRbase 21.0, and TargetScan 8.0. The interaction network map was constructed using miRNet 2.0 by integrating microarray data with microRNA databases.

### 2.14. Statistical Analysis

Statistical analysis was performed using GraphPad Prism 8. Data are presented as the mean ± s.d unless indicated otherwise, and *p*  <  0.05 was considered statistically significant. For all representative images, results were reproduced at least three times in independent experiments. For all the quantitative data, the statistical test used is indicated in the legends. A statistical ‘decision tree’ is provided in Appendix A. Heatmaps of the Z-transformed gene expression level of mRNA microarray were created using Python 3.7 Seaborne 0.9.0 (Micheal Waskom, NY, USA). Bubble-weighted plots with calculated q-values were created using the Python 3.6.2 circlize library.

## 3. Results

### 3.1. Transcriptomic Profiling Identified Differentially Regulated miRNAs, EMT, and Drug-Resistant Genes in the Rb Tumor Subtype

To obtain a broad view of miRNA regulation in Rb tumors, we first investigated the miRNA profile using a microarray. We performed a miRNA expression microarray in a primary Rb cohort (Table 1) comprising enucleated tumor tissues of five advanced (defined by AJCC staging- cT3 [22], IIRC- group E [23] and four non-advanced (defined by AJCC staging- cT2, IIRC- group D) subjects. Two age-matched pediatric retinas (age range from 2–3 months) with no ocular complications were used as controls. We identified sixteen distinct differentially regulated miRNAs that were unique to Rb tumors (*p* < 0.05, FC > 2) compared with the pediatric retina (Figure 1A). Notably, miR-181a-5p and miR-3653 were significantly downregulated in advanced Rb compared with non-advanced Rb tumors (Figure 1B). We applied KEGG pathway enrichment analysis to the miRNAs data obtained using the microarray and identified miRNA-regulating genes belonging to the cell cycle pathway, the EMT program, drug resistance, and pathways in cancer (Figure 1C). We performed RT-PCR validation experiments in a secondary cohort (Appendix A) comprising eight Rb tumor tissues (four advanced, four non-advanced) and four pediatric retina controls; this confirmed the downregulation of miR-181a-5p in Rb tumors, with significant downregulation in advanced subjects (*p* < 0.001) (Figure 1D). RT-PCR quantification of miR-331-3p, miR-574-5p, and miR-1290 in advanced and non-advanced Rb tumors corroborated with the expression profiles identified in the miRNA microarray (Appendix A). The findings prompted us to elucidate the EMT and drug resistance signatures in advanced and non-advanced Rb tumors. We performed total mRNA profiling using a gene expression microarray in the primary Rb cohort. We identified differentially regulated EMT and drug resistance genes in advanced (Figure 1E) and non-advanced Rb tumors (Figure 1F) compared with pediatric controls (*p* < 0.05, FC > 2). Notably, EMT transcription factors, such as *ZEB1* (FC = 92, *p* < 0.05) and *SNAI2* (FC = 5.57, *p* < 0.05), and drug resistance genes, such as *ABCB1* (MDR1) (FC = 5.84, *p* < 0.05) and *CTSL* (cathepsin L) (FC = 20.03, *p* < 0.05), were significantly upregulated in advanced tumors (Figure 1G). However, *ZEB1* (FC = 77.2, *p* < 0.05), *SNAI2* (FC = 3.32, *p* < 0.05), *ABCB1* (FC = 4.4, *p* < 0.05), and *CTSL* (FC = −3.8, *p* < 0.05) expressions were significantly downregulated in non-advanced Rb tumors. RT-PCR validations of these genes in a secondary cohort confirmed the findings of the microarray (Figure 1H–K).

### 3.2. Validation of the Epithelial-to-Mesenchymal Transition (EMT) and Chemo-Drug Resistance Proteins in Rb Tumors and Their Interaction with miR-181a-5p

Immunofluorescence analysis of FFPE specimens of Rb tumors detected strong ZEB1 and cathepsin L positivity in advanced Rb tumor tissues compared with non-advanced Rb (Figure 2A–C). However, we observed cathepsin L positivity in the photoreceptor layers of control tissues, indicating its lysosomal functions in the retina tissues [24], unlike its metastatic potential in cancers [25]. Notably, the advanced Rb tumors demonstrated a cadherin-switching phenotype with high expression of N-cadherin and low expression of E-cadherin in the tumor tissues (Figure 2D–F), which is suggestive of EMT dissemination [26]. RT-PCR validations displayed significant downregulation of EMT adhesion genes *CDH1* (E-cadherin) in advanced and non-advanced tumors (*p* < 0.001) (Appendix A), while *CDH2* (N-cadherin) maintained an elevated expression profile in advanced tumors (*p* < 0.001) (Appendix A), corroborating with the protein signals detected in advanced tumors. We also detected MDR1 positivity in advanced tumors indicating therapeutic resistance, while MDR1 expressions were vague in non-advanced tumors and undetected in pediatric retina tissues (Appendix A). For comprehensive functional analysis of miRNAs, we developed a miRNA–target interaction network map using miRNet 2.0 by integrating microarray data with the microRNA databases miRwalk, miRbase, and TargetScan. Out of the sixteen miRNAs identified in the Rb tumor microarray, nine miRNAs were associated with regulating cancer-specific pathways, while five miRNAs strictly regulated EMT pathway genes in the interaction map. We identified miR-181a-5p as a regulator of EMT transcription factors, such as *ZEB1* and *SNAI2*, while miR-124-3p was identified as a regulator for EMT facilitators, such as *CDH1* (E-cadherin) and *CHD2* (N-cadherin) (Figure 2G). However, miR-3653 did not display any interaction with KEGG-identified enriched pathways (Figure 2G). We speculated that advanced tumors maintain a high expression of EMT and drug resistance genes due to the low expression of miR-181a-5p, thus promoting invasion and metastasis (Figure 2H,I).

### 3.3. Chemotherapy-Resistant Rb Cells Conferred a High EMT Program and Metastasis

Initial regression of the Rb tumor is followed by an orbital relapse [27] or recurrence of more aggressive chemo-resistant tumors composed of tumor cells with a much higher tumor-initiating ability than the original tumor [28]. The EMT program through ZEB1 is known to drive cellular mobility and tumor dissemination in other cancer systems [29]; however, their role in EMT-driven drug resistance in Rb tumors is unknown. To extend our study of the consequences of RB1 downregulation in Rb tumors and its influence on miRNA and EMT signatures, we overexpressed RB1 in Rb-null Y79 and WERI-Rb1 cells. In real-time gene expression assays, we found miR-181a-5p to be significantly upregulated in the presence of Rb compared with Rb-null cells (*p* = 0.002) (Figure 3A and Appendix A). Rb overexpression decreased key EMT factors, such as ZEB1, Slug, N-cadherin, and drug-resistant MDR1 proteins, in the immunoblot (Figure 3B). However, Rb overexpression increased E-cadherin expression, indicating a halt in the mesenchymal transition (Figure 3B). These findings strongly suggest EMT as a modulator of mesenchymal phenotype and drug resistance to further promote invasion and migration (Figure 3C). To expose the mechanism, we developed Y79 cells that were resistant to topotecan and carboplatin and WERI-Rb1 cells resistant to topotecan by exposing them to increasing concentrations of the drugs for 3 weeks (Figure 3C). After each week, the surviving cells that reached >60% confluency were passaged in fresh media with an increased concentration of topotecan or carboplatin. The procedure was performed repeatedly until the cells display low sensitivity to IC_50_ doses of topotecan or carboplatin (Figure 3D,E and Appendix A), reduced DNA damage repair process defined by low λH2A.X foci count under an IC_50_ dose therapy for 48 h (Appendix A), a shift in IC_50_ values of topotecan and carboplatin (Appendix A), and a high surface expression of MDR1 proteins (Figure 3F–H, Appendix A), marking a resistant phenotype. We developed tumor spheroids for parental and resistant Y79 cells, and we observed high ZEB1 and cathepsin L expression in resistant spheroids using immunofluorescence. However, parental spheroids displayed strong ZEB1 expression and no cathepsin L expression (Figure 3I). We further confirmed the findings using RT-PCR that revealed a mesenchymal transition trend for resistant lines compared with parental lines (Figure 3J–M, Appendix A). Likewise, using a transwell assay, we detected an increase in invasion and migration properties of resistant Y79 and WERI-Rb1 cells under high-dose topotecan (100 nM) therapy (Figure 3N–P and Appendix A). In line with the above results, in human Rb tumors, miR-181a-5p was significantly downregulated in EMT-high/drug-resistant advanced tumors. All these data pointed to miR-181a-5p as a previously unrecognized negative regulator of the EMT program and drug resistance mechanism that possibly influences tumor metastasis (Appendix A).

### 3.4. Resistant Cells Elicited a Transition through ZEB1 and Resistance through Cathepsin L

To identify the critical downstream signaling pathways that regulate EMT and chemo-resistance, we focused on the TGFβ pathway, which was the most overrepresented among the advanced Rb tumors in the microarray analysis (Appendix A). To substantiate this finding, using a Western blot, we found an increased expression of phospho-SMAD2 in resistant Y79 compared with parental (Figure 4A). The resistant cells showed more pronounced expression of EMT markers, such as ZEB1, Slug, and N-cadherin, and drug-resistant markers, such as MDR1 and cathepsin L (Figure 4A), mimicking an advanced tumor signaling circuit. In agreement with a previous report stating that retinoblastoma cells lack functional TGFβ receptors I and II [30] (Figure 4B and Appendix A), we identified that ACVRC1 receptors, which is a member of the TGFβ family, could accelerate SMAD2 activations in advanced retinoblastoma [31] (Appendix A). In real-time gene expression assays, we confirmed our findings in resistant cells that showed an increase in the expression of *ACVRC1* transcript, pointing out the role of TGFβ signaling in advanced tumors (Figure 4C and Appendix A). To see whether TGFβ modulation affected the resistance phenotype, we used TGFβ ligand activation (10 ng) and TGFβ inhibitor (50 µM SB43152) for 48 h in parental and resistant lines and assessed the changes in EMT and drug resistance markers. Using immunofluorescence, we found enhanced levels of phospho-SMAD2 and cathepsin L levels upon TGFβ activation in resistant cells (Figure 4D and Appendix A), while TFGβ-activated parental cells showed a slight increase in phospho-SMAD2 and cathepsin L levels compared with the controls. Likewise, TGFβ activation also increased the ZEB1 expression in resistant and parental cells compared with the controls (Appendix A). TGFβ inhibition in parental lines shows a complete reduction in phospho-SMAD2, ZEB1, and cathepsin L proteins; however, resistant cells upon TGFβ inhibition showed a partial reduction in cathepsin L and ZEB1 in the nucleus (Figure 4D and Appendix A). Consistently, we observed lower levels of ZEB1, Slug, and cathepsin L proteins upon TGFβ/phospho-SMAD2 inhibition in resistant lines using Western blotting (Figure 4E). In contrast to the resistant phenotype, TGFβ/SMAD2 inhibition drastically reduced ZEB1, Slug, and depleted cathepsin L proteins in parental lines. TGFβ inhibition showed significant downregulation of *SMAD2* (Appendix A) and *ZEB1* genes in the resistant lines (Figure 4F and Appendix A), while TGFβ inhibition did not affect *SNAI2* and *CTSL* (cathepsin L) expressions in resistant cells (Figure 4G,H and Appendix A). Thus, TGFβ transcriptionally regulated *ZEB1* but not *SNAI2* and *CTSL* in resistant lines. Using promoter binding analysis, we found that the ZEB1 promoter had direct binding sites for *SMAD2* that were closer and within the transcription start site (TSS), which was suggestive of strong transcriptional activation of ZEB1 (Appendix A). However, *SMAD2* binding sites in the SNAI2 promoter were relatively far off from the TSS (Appendix A), confirming its poor sensitivity to TGFβ inhibitors. Notably, *SNAI2* also had strong promoter binding sites in the CTSL promoter, and thus, it is possible that the transcriptional activation of *CTSL* was mediated via *SNAI2* and not *SMAD2* or *ZEB1* (Appendix A). In support of our results, we observed enhanced nuclear localization of cathepsin L in resistant lines indicating its transcriptional activity independent of TGFβ/SMAD2 signals (Appendix A). We speculated that resistant cells have nuclear localization of CTSL due to the lack of steffin B (*CSTB*) [32], as evidenced in an advanced tumor microarray (Figure 2A). This indicated that ZEB1 triggers EMT through TGFβ and the activated EMT program through SNAI2 regulates CTSL-mediated chemoresistance in advanced Rb tumors (Figure 4I). Hence, identifying a common regulator, such as miR-181a-5p, that governs the mechanisms of both the transition and resistance is both reasonable and promising for the effective management of metastatic dissemination.

### 3.5. Resistance Depletion by miR-181a-5p Conferred Sensitivity to Chemotherapy

To understand how miR-181a-5p affected the EMT and chemoresistance mechanism in resistant Y79 lines, using Western blotting, we focused particularly on the ZEB1, Slug, and cathepsin L proteins. Using the bioinformatic tools miRwalk and Targetscan, we also predicted the binding sites of miR-181a-5p in the ZEB1 and SNAI2 3′ UTR regions (Appendix A). In contrast to mimic control, the overexpression of miR-181a-5p drastically reduced the ZEB1, Slug, and cathepsin L protein levels (Figure 5A). However, miR-181a-5p inhibition showed an opposing result, mimicking an EMT with a highly drug-resistant advanced tumor phenotype (Figure 5A). We accordingly observed changes in the MDR1 surface expression in resistant cells overexpressed with miR-181a-5p (Figure 5B,C and Appendix A) that were partly explainable by changes in the protein levels of Slug and cathepsin L in the immunoblot. Notably, miR-181a-5p overexpression reduced the cell proliferation (Figure 5D), invasion, and migration of resistant cells (Figure 4E–G and Appendix A). However, miR-181a-5p inhibition showed contrary results by increasing all cancer hallmarks in the resistant lines (Figure 5D–G and Appendix A). These findings led us to hypothesize that miR-181a-5p-overexpressed resistant lines might be particularly sensitive to low-dose chemotherapy. To test this, we first compared the response of miR-181a-5p-modulated resistant lines to the IC_50_ dose of topotecan (parental Y79 and WERI-Rb1 IC_50_ = 10 nM) for 96 h. Following topotecan treatment, the miR-181a-5p-overexpressed resistant lines did not show any significant response at 24 h and 48 h (Figure 5H,I), while they showed increased sensitivity and low survival to the topotecan IC_50_ by 72 h (Figure 5J and Appendix A) and 96 h (Figure 5K), confirming the efficacy of miR-181a-5p. Together the results suggested that the miR-181a-5p played a major role in the depletion of EMT and resistant phenotype that further sensitized the cells to low-dose chemotherapy.

## 4. Discussion

The present study identified miR-181a-5p as a previously unrecognized regulator of EMT transcription factors and chemotherapy resistance. While the study focused on intraocular advanced and non-advanced retinoblastoma tumors, our findings can be extended to other cancer systems that have persistent EMT-associated chemotherapy resistance. We found miR-181a-5p to be significantly downregulated in advanced Rb tumors (FC = −62.96, *p* < 0.05) and provided evidence that the mesenchymal transition and chemoresistance in tumors are likely sensitive to chemotherapy when miR-181a-5p is complemented.

The tumor-promoting [33,34] and tumor-suppressing [35,36] roles were reported for miR-181-5p. However, here we report the downregulation of miR-181a-5p in retinoblastoma tumor tissues from patients when compared with healthy pediatric retinae. Discrepancies in the role of miR-181-5p existed in our study due to the limited cohort sizes of Rb tumors and the use of pediatric retinae as controls. The miRNA 181a-5p also plays a functional role in the retina [37], hence the consistent low expression of miR-181a-5p expression in advanced and non-advanced retinal tumors was predictable. Our work not only provides evidence for the tumor suppressor function of miR-181a-5p but also highlights the need to better dissect the dual role of miR-181a-5p in various cancers.

The control of EMT and chemotherapy resistance by miR-181a-5p evidenced here could provide an explanation for the apparent complex roles of miR-181a-5p in advanced tumors. Recent evidence indicates that EMT occurs through intermediate states rather than being a binary process [38] and is partially reactivated in various cancers [39]. We propose that dynamic fluctuations in miR-181a-5p levels in tumor and control tissues and between different stages of Rb progression may contribute to the context-dependent EMT plasticity from cancer initiation to metastasis. Differences in EMT and drug resistance transcripts between control and tumor tissues and different stages of Rb further contribute to this complexity, as miR-181a-5p controls the EMT in a tissue- and function-specific manner. In this study, advanced Rb tumors showed increased expression of EMT signatures (*ZEB1*, FC = 92, *p* < 0.05; *SNAI2*, FC = 5.57, *p* < 0.05), which was a consequence of miR-181a-5p downregulation, as an EMT trigger. We proposed that the EMT genes, such as *ZEB1* and *SNAI2*, acquired transcript stability in advanced tumors, at least partly due to reduced miR-181a-5p-based degradation [40]. Furthermore, we identified chemotherapy resistance pathway genes, such as *ABCB1* (MDR1) [41] and *CTSL* (cathepsin L) [42], as EMT targets in Rb tumors. Likewise, the miR-181a-5p clusters located on chromosome 1 are known to repress E2F transcription factors [43], G1/S cell cycle regulators [44], and proto-oncogenes [45]. We found that miR-181a-5p negatively controls EMT and chemoresistance through the regulation of SNAI2 and CTSL transcripts in vitro. Collectively, our work reveals an emerging and intriguing feature of miR-181a-5p and its association with a variety of distinct signaling pathways in Rb tumors.

We identified the balance between EMT-driven metastasis in Rb tumors to be influenced by secreted cytokine TGFβ in the tumor microenvironment, which is a known promoter of EMT in Rb-depleted tumors [46]. Previous studies highlighted the lack of canonical TGFβ receptors in Rb cells [30], and in agreement with recent reports [31], we showed that TGFβ signals act through ACVRC1 receptors and activate SMAD2/3 effectors in Rb. Mechanistically, we showed that the presence of the TGFβ ligand mediates a mesenchymal shift in Rb cells and is associated with enhanced migration and invasion capacity. Conversely, treatment with a TGFβ signaling inhibitor reduced ZEB1 and SNAI2 levels and prevented the acquisition of mesenchymal marker expression and morphological features, thus linking mesenchymal differentiation in Rb with enhanced tumor cell invasion through the TGFβ/ZEB1/SNAI2 axis. Interestingly, and in line with our observations of the importance of miRNA, EMT, and chemoresistance, a recent report highlights that the miR-200c-ZEB1 feedback loop is involved in the invasion, migration, and chemoresistance in advanced glioblastoma tumors [47]. We found that TGFβ signals can drive an EMT program in Rb-/- cells, while they do not necessarily lead to chemoresistance. We observed that Rb cells acquire chemotherapy resistance through an enhanced EMT program that is orchestrated by SNAI2 by regulating CTSL. This concept was further supported by our observations with the ectopic expression of miR-181a-5p, which represses SNAI2 at its 3′ UTR region and targets the SNAI2-CTSL signaling cascade, inhibiting transition and resistance.

The tumor suppressor properties of miRNA in various cancers have prompted the development of various potent inhibitors of pharmacological targeting in clinical settings [48], but our study was limited to in vitro models and lacks investigations in animal models. On the other hand, our findings on miR-181a-5p raise hopes for therapeutic strategies for the management of advanced Rb tumors.

## 5. Conclusions

We explored the possible role of EMT and drug resistance in advanced Rb tumors and highlights the role of miR-181a-5p in EMT- and chemoresistance-related gene expression and drug sensitivity in retinoblastoma. In conclusion, our data revealed a mechanistic link between EMT and chemoresistance in Rb tumors, which was mediated by miR-181-5p. Thus, we identified miR-181a-5p as a potential target to control the tumor EMT program and development of chemoresistance, which is an encouraging prospect for cancer research and therapy

## Figures and Tables

**Figure 1 cancers-14-05124-f001:**
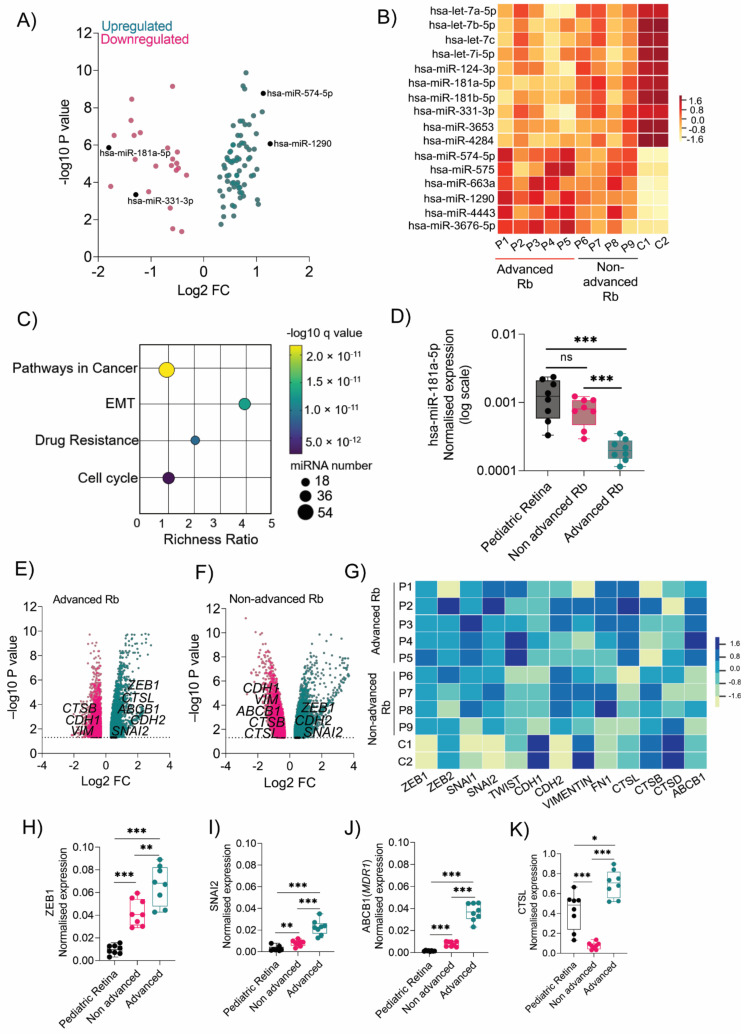
Transcriptomic profiling identified differentially regulated miRNAs, EMTs, and drug resistance genes in Rb tumor subtypes. (**A**) Volcano plot showing differentially regulated miRNAs in Rb subjects (*n* = 9) compared with the pediatric retinas (*n* = 2) identified using a microarray. (**B**) Heatmap showing differential expression of miRNAs in 9 Rb subjects and 2 pediatric controls identified using a microarray. (**C**) Bubble scatter plot showing the top enriched KEGG pathways regulated by miRNAs in Rb tumors. (**D**) RT-PCR results showing the normalized expression of miR-181a-5p in the control retinas (*n* = 4), advanced Rb (*n* = 4), and non-advanced Rb (*n* = 4). Volcano plot showing the differentially regulated EMT and chemotherapy resistance genes identified using a microarray in (**E**) advanced Rb tumors and (**F**) non-advanced Rb tumors. (**G**) Heatmap showing the expression of EMT and chemotherapy resistance genes in 9 Rb subjects and 2 pediatric controls. RT-PCR showing the normalized expression of (**H**) *ZEB1*, (**I**) *SNAI2*, (**J**) *ABCB1*, and (**K**) *CTSL* in control pediatric retinas (*n* = 4), advanced Rb tumors (*n* = 4), and non-advanced Rb tumors (*n* = 4). Values represent the mean ± s.d. Two-tailed Mann–Whitney was used for the statistical analysis. * *p* < 0.05, ** *p* < 0.01, *** *p* < 0.001. ’ns’ represents no statistically significant difference between the means of two variables.

**Figure 2 cancers-14-05124-f002:**
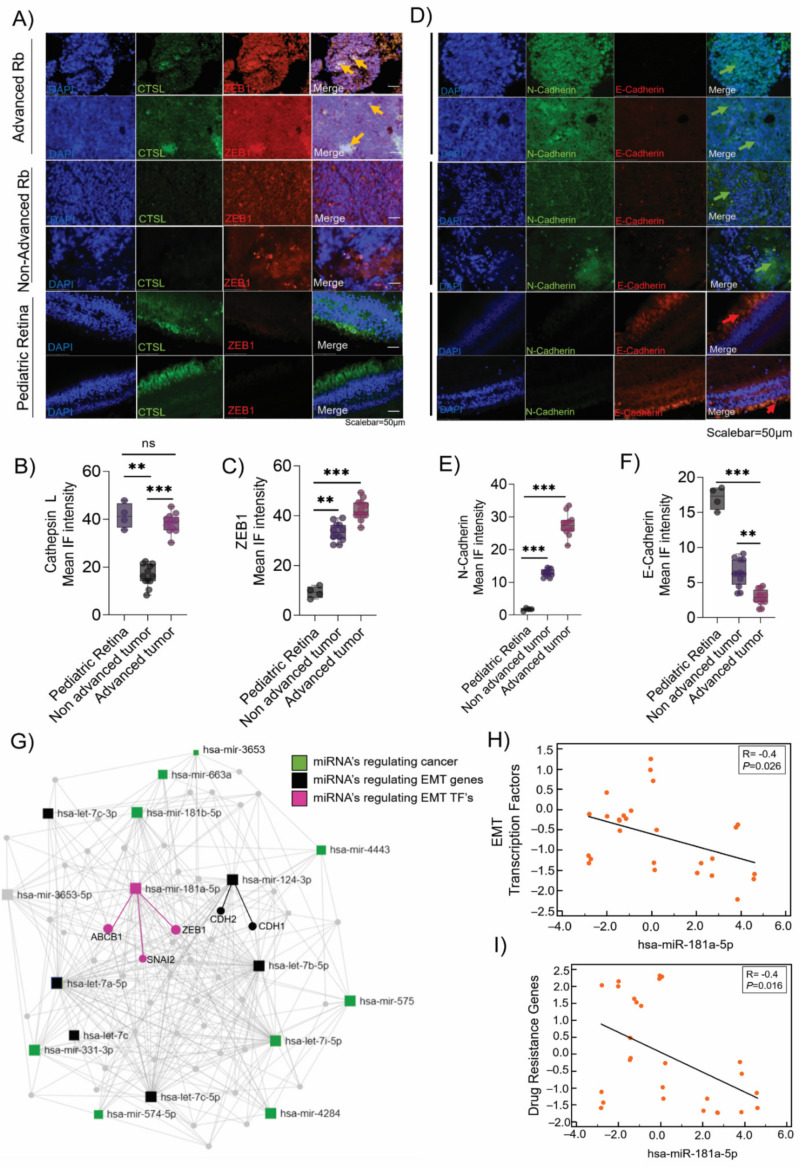
Validation of epithelial-to-mesenchymal transition (EMT) and chemo-drug resistance genes in Rb tumors and their correlation with miR-181a-5p. Immunofluorescence results showing the expression of (**A**) ZEB1 and CTSL. The IF mean intensity of (**B**) ZEB1 and (**C**) cathepsin L staining in advanced tumors (*n* = 12), non-advanced tumors (*n* = 12), and control pediatric retinas (*n* = 4). Immunofluorescence results showing the expression of (**D**) N-cadherin and E-cadherin, IF mean intensity of (**E**) N-cadherin and (**F**) E-cadherin in advanced tumors (*n* = 12), non-advanced tumors (*n* = 12), and control pediatric retina tissues (*n* = 4). Scale bar: 50 µm. (**G**) Network map showing the predicted interaction of miRNA–mRNA targets using miRNet. Correlation plot showing the (**H**) negative correlation of EMT genes (*ZEB1, SNAI2,* and *TWIST*) with miR-181a-5p in Rb tumors and (**I**) negative correlation of drug resistance genes (*MDR1, MRP1*, and *CTSL*) with miR-181a-5p in Rb tumors. Values represent the mean ± s.d. Two-tailed Mann–Whitney was used for statistical analysis. ** *p* < 0.01, *** *p* < 0.001. ’ns’ represents no statistically significant difference between the means of two variables.

**Figure 3 cancers-14-05124-f003:**
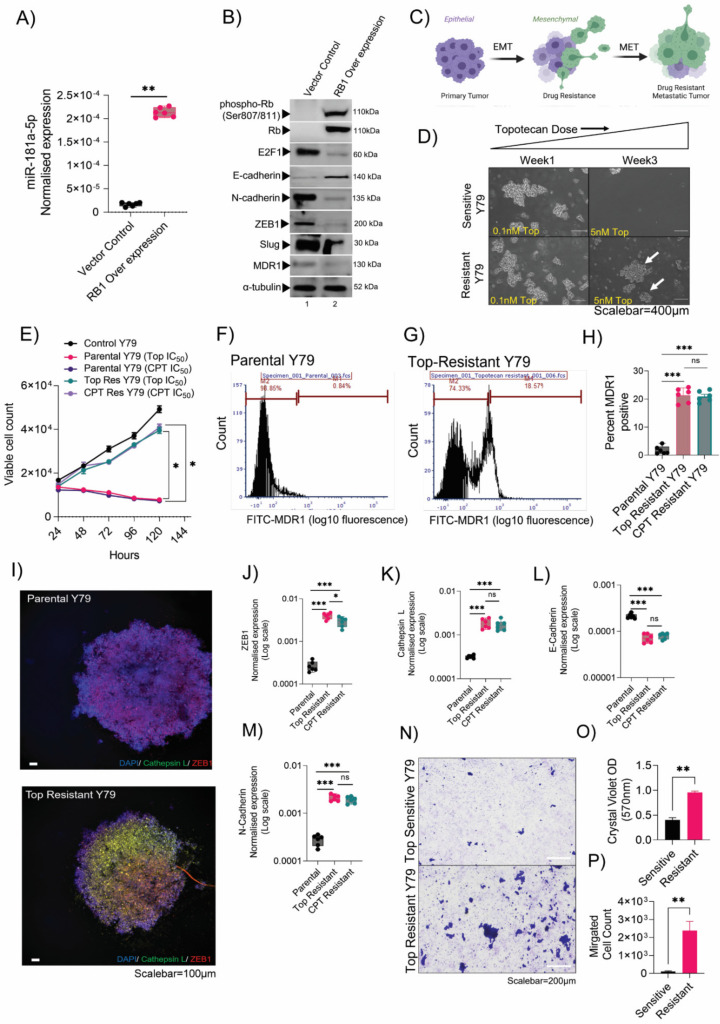
Chemotherapy-resistant Rb cells conferred a high-EMT program and metastasis. (**A**) RT-PCR showing normalized expression of miR-181a-5p in vector control (*RB1*-null) and *RB1*-overexpressed Y79 retinoblastoma cells. (**B**) Immunoblot showing the expression of the EMT and chemo-resistant markers in vector control (*RB1* null) and *RB1*-overexpressed Y79 cells. (**C**) Schematic showing the EMT program and drug resistance induction in metastatic tumors. (**D**) Phase contrast microscopy images showing the morphology of parental and resistant Y79 cells under increasing doses of topotecan treatments from week 1 to week 3. Scale bar: 400 µm. (**E**) Trypan blue cell count of parental, topotecan-resistant, and carboplatin-resistant Y79 cells for 24 h, 48 h, 72 h, and 96 h. MDR1 surface expression analysis in (**F**) parental and (**G**) topotecan-resistant Y79 cells by flow cytometry. (**H**) Bar graph showing the percentage of cells positive for MDR1 surface expression in parental, topotecan-resistant, and carboplatin-resistant Y79 cells. (**I**) Parental and resistant Y79 spheroids showing the expression of ZEB1 and cathepsin L. Scale bar: 100 µm. RT-PCR showing expression of (**J**) ZEB1, (**K**) cathepsin L, (**L**) E-cadherin, and (**M**) N-cadherin in parental, topotecan-resistant, and carboplatin-resistant Y79 cells. (**N**) Transwell invasion and migration assay to assess the migratory capacity of resistant cells compared to sensitive cells under a 10 nM topotecan treatment for 48 h. (**O**) Crystal violet OD reading at 570 nm to assess invasiveness. (**P**) Trypan blue count to assess migrated cells in the lower compartment of the transwell chamber. Two-tailed Student’s *t*-test (for 2 groups) and one-way ANOVA with Dunnett’s multiple comparisons tests (for >2 groups) were used for the statistical analysis. * *p* < 0.05, ** *p* < 0.01, *** *p* < 0.001. ‘ns’ represents no statistically significant difference between the means of two variables.

**Figure 4 cancers-14-05124-f004:**
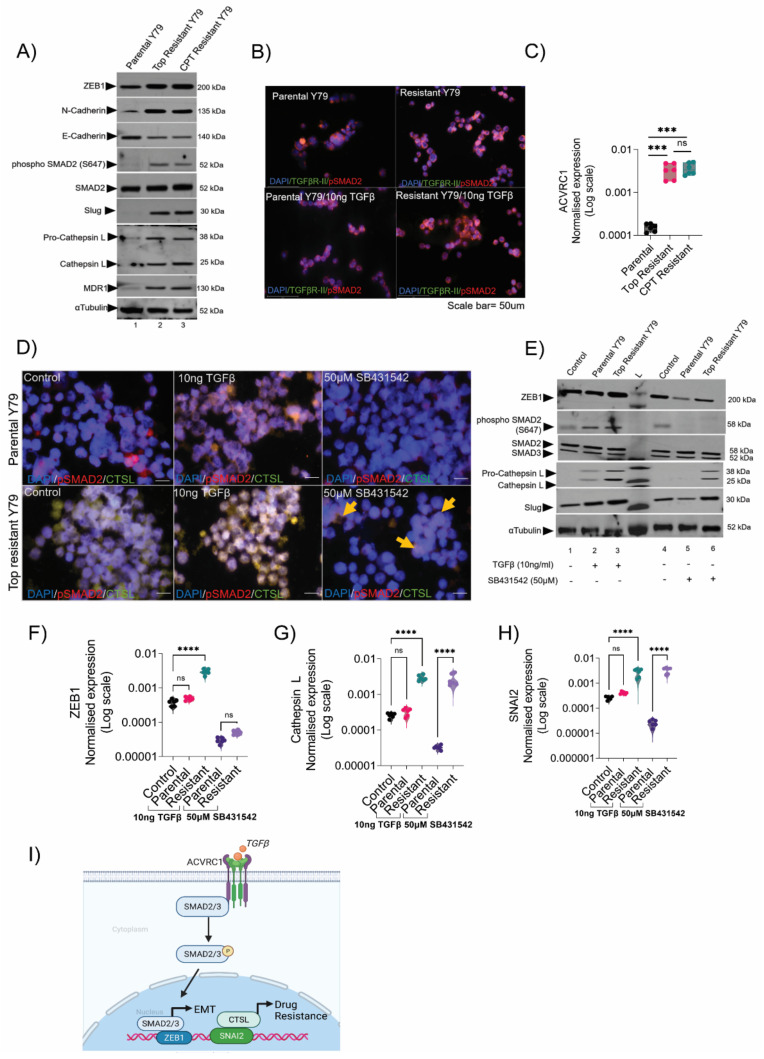
Resistant cells elicited a transition through ZEB1 and resistance through cathepsin L. (**A**) Immunoblot showing the expression of EMT and drug resistance markers in parental, topotecan-resistant, and carboplatin-resistant Y79 cells. Uncropped immunoblots are provided in Appendix A. (**B**) Immunofluorescence showing the expression of TGFβ R-II and phospho-SMAD2 in parental and topotecan-resistant Y79 cells with and without TGFβ induction. Scale bar: 50 µm. (**C**) RT-PCR results showing the expression of *ACVRC1* in parental, topotecan-resistant, and carboplatin-resistant cells. (**D**) Immunofluorescence showing the expression of phospho-SMAD2 and cathepsin L (CTSL) upon TGFβ induction (10 ng for 48 h) and TGFβ inhibition (50 µM SB431542 for 48 h) in parental and topotecan-resistant Y79 cells. Scale bar: 50 µm. (**E**) Immunoblot showing the differential regulation of proteins belonging to the EMT and drug resistance pathway upon TGFβ induction and inhibition for 48 h. Uncropped immunoblots are provided in Appendix A. RT-PCR results show the normalized expression of (**F**) *ZEB1*, (**G**) *SNAI2*, and (**H**) *CTSL* (cathepsin L) upon TGFβ induction and inhibition for 48 h. (**I**) Schematic showing the novel regulation of EMT and drug resistance mechanism in Rb tumors. Two-tailed Student’s *t*-test (for 2 groups) and one-way ANOVA with Dunnett’s multiple comparisons tests (for >2 groups) were used for the statistical analysis. *** *p* < 0.001, **** *p* < 0.0001. ‘ns’ represents no statistically significant difference between the means of two variables.

**Figure 5 cancers-14-05124-f005:**
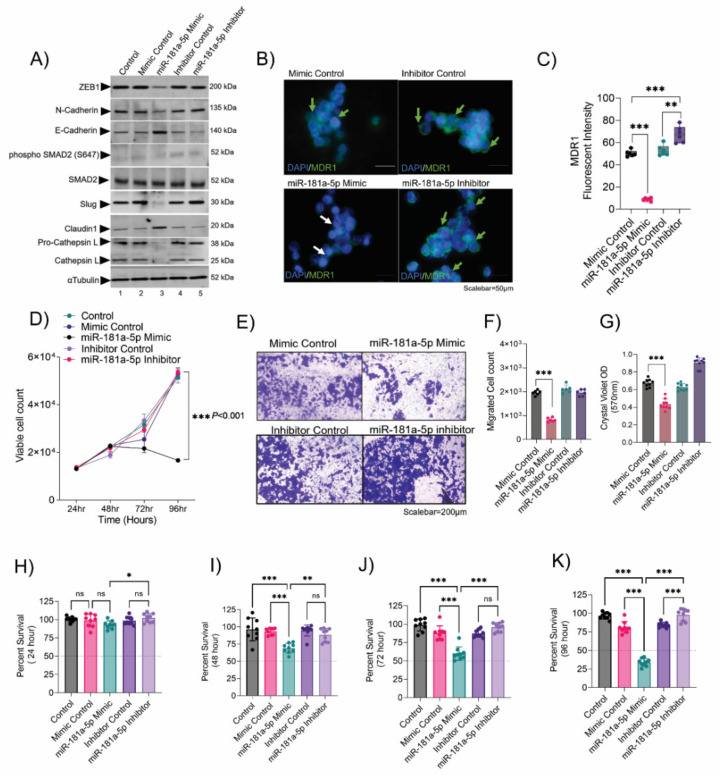
Resistance depletion by miR-181a-5p conferred sensitivity to chemotherapy. (**A**) Immunoblot showing the expression of key EMT factors and drug resistance markers upon miR-181a-5p overexpression and inhibition in topotecan-resistant Y79 cells. Uncropped immunoblots are provided in Appendix A. (**B**) Immunofluorescence showing the MDR1 surface expression in topotecan-resistant Y79 cells upon miR-181a-5p overexpression and inhibition. Scale bar: 50 µm. (**C**) Bar graphs showing the MDR1 fluorescent intensity in topotecan-resistant Y79 cells upon miR-181a-5p overexpression and inhibition. (**D**) Trypan blue cell count showing the proliferation of topotecan-resistant cells at 24 h, 48 h, 72 h, and 96 h upon miR-181a-5p overexpression and inhibition. (**E**) Transwell invasion and migration assay to assess the invasive and migratory capacity of topotecan-resistant cells upon miR-181a-5p overexpression and inhibition. (**F**) Crystal violet OD measurement at 570 nm to assess the invasiveness of resistant Y79 cells. (**G**) Trypan blue cell count showing the migrated cells in the lower compartment of the transwell chamber. Chemosensitivity of miR-181a-5p modulated topotecan-resistant Y79 cells upon 10 nM topotecan treatment for (**H**) 24 h, (**I**) 48 h, (**J**) 72 h, and (**K**) 96 h. The control represents untreated topotecan-resistant Y79 cells. Two-tailed Student’s *t*-test (for 2< group) and one-way ANOVA with Dunnett’s multiple comparisons tests (for >2 groups) were used for the statistical analysis. * *p* < 0.05, ** *p* < 0.01, *** *p* < 0.001. ‘ns’ represents no statistically significant difference between the means of two variables.

**Table 1 cancers-14-05124-t001:** Clinical and histopathological details of the samples.

ID	Sex	Laterality	Age at Presentation	Clinical Risk	IIRC Group	AJCC Staging
P1	M	Bilateral	15 months	Advanced	Group E	cT3b
P2	F	Unilateral	20 months	Advanced	Group E	cT3b
P3	M	Unilateral	24 months	Advanced	Group E	cT3a
P4	F	Bilateral	4 months	Advanced	Group E	cT3b
P5	M	Bilateral	30 months	Advanced	Group E	cT3b
P6	F	Bilateral	21 months	Non-advanced	Group D	cT2b
P7	F	Unilateral	28 months	Non-advanced	Group D	cT2b
P8	M	Unilateral	20 months	Non-advanced	Group D	cT2b
P9	M	Unilateral	21 months	Non-advanced	Group D	cT2a
Control 1	F	NA	3 months	Cardiac arrest (no ocular complications)		
Control 2	F	NA	2 months	Multiple organ dysfunction (no ocular complications)		

**Table 2 cancers-14-05124-t002:** Details of the qPCR primers used in the study.

Gene	Forward Primer	Reverse Primer	Tm (F/R)
ZEB1	GCCTCCTATAGCTCACACATAAG	TGCTGGAAGAGACGGTGAA	56.67/56.8
SNAI2	GTGATTATTTCCCCGTATCTCTAT	TCAATGGCATGGGGTCTGA	55.6/60.2
CDH1 (E-cadherin)	GAAGGTGACAGAGCCTCTGGAT	GATCGGTTACCGTGATCAA	57.2/58.4
CDH2 (N-cadherin)	CGAGCCGCCTGCGCTGCCAC	CGCTGCTCTCCGCTCCCCGC	56.5/57.3
ACVRC1	AGGAGTTTCGACCCCAGTAA	GTAGCACTTACCGTAGCACC	57.9/58.2
CTSL	AGGCCTGGACTCTGAGGAAT	AGCCGGTGTCATTAGCAACA	57.8/57
SMAD2	CCGCCAGTTGTGAAGAGACT	CTGCCCATTCTGCTCTCCTC	59.9/60.1
ABCB1	GAGCAGTCATCTGTGGTCTT	CCCCTTCAAGATCCATTCCG	57.2/58.0
β-Actin	TCCCTGGAGAAGAGCTACGA	AGGAAGGAAGGCTGGAAGAG	56.9/55.2

**Table 3 cancers-14-05124-t003:** Details of miRNA primers used in the study.

S. No.	Systematic Name	Regulation	Mirbase Accession No	Active Sequence
1	hsa-miR-331-3p	Down	MIMAT0000760	TTCTAGGATAGGCCCAGGG
2	hsa-miR-181a-5p	Down	MIMAT0000256	ACTCACCGACAGCGT
3	hsa-miR-574-5p	Up	MIMAT0004795	ACACACTCACACACACAC
4	hsa-miR-1290	Up	MIMAT0005880	TCCCTGATCCAAAAATCC

## Data Availability

mRNA (GSE208143) and miRNA (GSE208677) microarray data were submitted to the NCBI GEO database. All data and source files will be available on request.

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
