# Peer review of "Enhanced Epithelial-to-Mesenchymal Transition and Chemoresistance in Advanced Retinoblastoma Tumors Is Driven by miR-181a"

_cancers, 2022, doi:10.3390/cancers14205124_

Round 1

Reviewer 1 Report

The authors of the paper titled 'Enhanced epithelial to mesenchymal transition and chemo-resistance in advanced Retinoblastoma tumours is driven by miR-181a' have discovered miR-181a-5p as a therapeutically useful target for drug-resistant EMT-associated tumours, which stops their invasion, migration and sensitises them to low doses of chemotherapeutic drugs (The authors studied advanced retinoblastoma tumours, but their findings can also be extended to other tumour systems, i.e. those with persistent resistance to chemotherapy associated with EMT). After a detailed review of the text of the publication, the results described in the paper, an assessment of the selection and range of research methods, and verification of the statistical analyses undertaken, I conclude that the paper fully and already at this stage merits publication in the journal Cancers. I recommend this work for acceptance for publication. To the authors, I offer my sincere congratulations and wish them all the best.

Reviewer 2 Report

Babu and colleagues present an interesting paper profiling mRNA and microRNA in primary specimens of advanced and non-advanced retinoblastoma as well as normal pediatric retina. This work in primary tissue is complemented by analyses of 2 retinoblastoma cell lines.

-          Y79 and WERI-Rb1 cells were treated with 10-fold increases of carboplatin and topotecan for 3 weeks to make drug resistant. Can the authors clarify as to whether there was a ‘wash-out’ phase where these cell lines were grown without drugs for a period of time, followed by re-testing of drug resistance to ascertain that the cell lines that were made were stably drug resistant?

-          While I imagine it is extremely difficult to obtain normal pediatric retina, the 2 samples obtained were much younger (children of 2 and 3 months of age) than the retinoblastoma samples (9 samples, mean age ~ 20 months, split into 2 groupings of advanced (n=5) and non-advanced (n=4)). Can the authors please comment on this in the context of results.  Having only two normal samples is a weakness of this study, although in the context of difficulty of attainment, I would request that the authors just mention this issue. Still, a secondary cohort is discussed (8 Rb tumours (4 of each advanced and non-advanced) and 4 pediatric retina controls. Can the authors please comment on why their primary study did not include some of these additional control retinas?

-          An advantage of this study is that miRNA and mRNA expression were determined on the same samples. This is a strong study design, but doesn’t exclude secondary effects. Can the authors please comment on this?

-          The authors use predictive programs such as Targetscan to suggest that miR-181a-5p is binding to and regulating ZEB1 and SNAI2 3’ UTR regions. No experiments are reported to confirm this (luciferase assays?). Can the authors please include these experiments or make a case for why they are not required?

Reviewer 3 Report

The authors identify miR-181a-5p as a therapeutically exploitable target for EMT-triggered drug-resistant cancers.  The authors need to confirm the following recommendations.

*In Table 2, why are some primers listed, but not used in the article or figure? For example, hsa-let-7c. In addition, the primers of some genes (ABCB1...) are used, but they are not listed in table 2. It is recommended to list them in full.
*The authors predicted the binding sites of miR-181a-5p in ZEB1 and SNAI2 3’ UTR regions. 
Why is there no experimental verification that miRNA can directly bind to 3’UTR of these genes?

Reviewer 4 Report

An interesting, complex study, difficult but with good results. The problems is the applicatio in practice for financial reasons. 

The conclusions must be improved!

Good job!
